# Synthesizing Moving People with 3D Control

## Abstract

In this paper, we present a diffusion model-based methodology for animating people from a single image for a given target 3D motion sequence. Our approach has two core components: a) learning priors about invisible parts of the human body and clothing, and b) rendering novel body poses with proper clothing and texture. For the first part, we learn an in-filling diffusion model to hallucinate unseen parts of a person given a single image. We train this model on texture map space, which makes it more sample-efficient since it is invariant to pose and viewpoint. Second, we develop a diffusion-based rendering pipeline, which is controlled by 3D human poses. This produces realistic renderings of novel poses of the person, including clothing, hair, and plausible in-filling of unseen regions. This disentangled approach allows our method to generate a sequence of images that are faithful to the target motion in the 3D pose and, to the input image in terms of visual similarity. In addition to that, the 3D control allows various synthetic camera trajectories to render a person. Our experiments show that our method is resilient in generating prolonged motions and varied challenging and complex poses compared to prior methods. Please check our anonymous demo for more details: link.

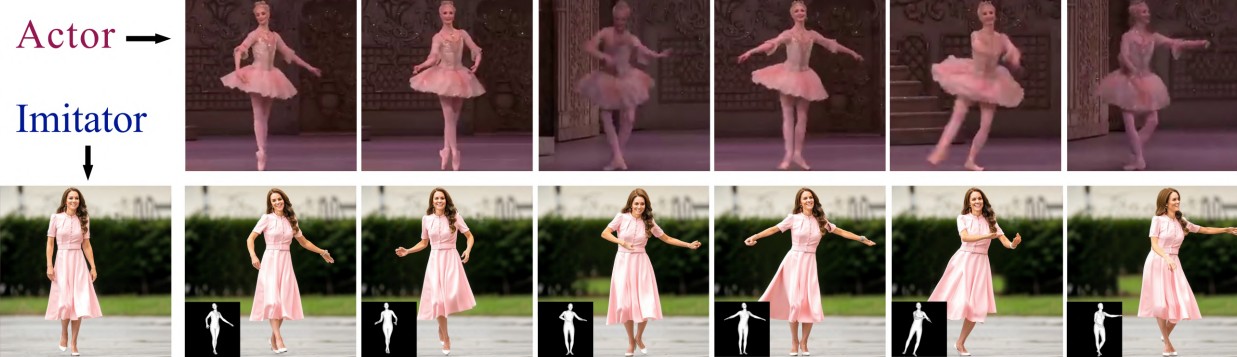

Figure 1: **The Imitation Game:** Given a video of a person **"The Actor"**, we want to transfer their motion to a new person **"The Imitator"**. In this figure, the first row shows a sequence of frames of the actor from a ballerina *Dance of the Sugar Plum Fairy*. The inset row shows the 3D poses extracted from this video. Now, given any single image of a new person **"The Imitator"**, our model can synthesize new renderings of the imitator to copy the actions of the actor in 3D. Please check more results in our anonymous demo.

## 1 Introduction

Given a random photo of a person, can we accurately animate that person to imitate someone else's action? This problem requires a deep understanding of how human poses change over time, learning priors about human appearance and clothing. For example, in Fig. 1, the **Actor** can do a diverse set of actions, from simple actions such as walking and running to more complex actions such as fighting and dancing. For the **Imitator**, learning a visual prior about their appearance and clothing is essential to animate them at different poses and viewpoints.

To tackle this problem, we propose **3DHM**, a diffusion framework (see Fig. 2) that synthesizes **3D H**uman **M**otions by completing a texture map from a single image and then rendering the 3D humans to imitate the actions of the actor.

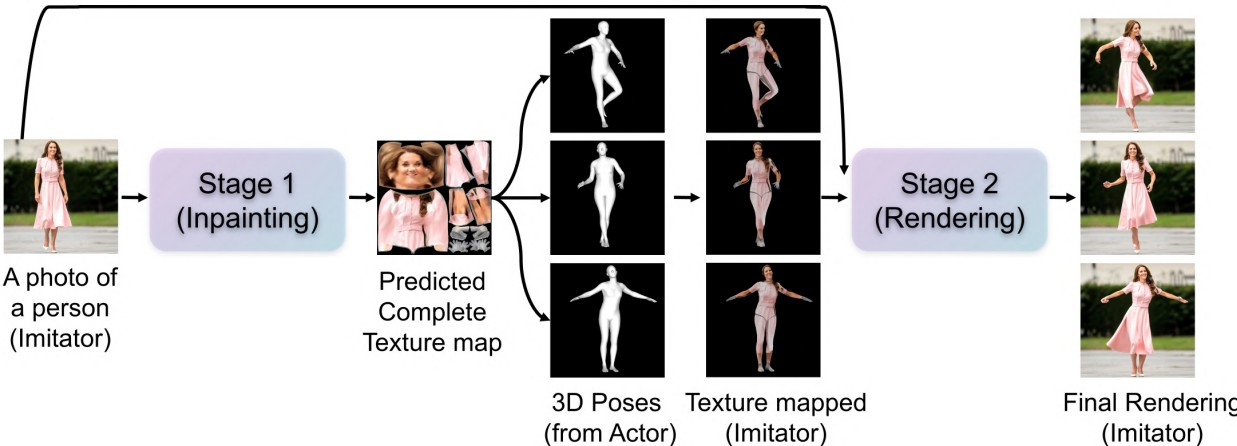

Figure 2: **Overview of 3DHM:** we show an overview of our model pipeline. Given an image of the imitator and a sequence of 3D poses from the actor, we first generate a complete full texture map of the imitator, which can be applied to the 3D pose sequences extracted from the actor to generate texture-mapped intermediate renderings of the imitator. Then we pass these intermediate renderings to the Stage-2 model to project the SMPL mesh rendering to more realistic renderings of real images.

We use state-of-the-art 3D human pose recovery model 4DHumans (Rajasegaran et al., 2022; Goel et al., 2023) for extracting motion signals of the actor, by reconstructing and tracking them over time. Once we have a motion signal in 3D, as a sequence of meshes, one would think we can simply re-texture them with the texture map of the imitator to get an intermediate rendering of the imitation task. However, this requires a complete texture map of the imitator. When given only a single view image of the imitator, we see only a part of their body, perhaps the front side, or the backside but never both sides. To get the complete texture map of the imitator from a single view image, we learn a diffusion model to in-fill the unseen regions of the texture map. This essentially learns a prior about human clothing and appearance. For example, a front-view image of a person wearing a blue shirt would usually have the same color at the back. With this complete texture map, now we can get an intermediate rendering of the imitator doing the actions of the actor. Intermediate rendering means, wrapping the texture map on top of the SMPL Loper et al. (2023) mesh to get a body-tight rendering of the imitator.

However, the SMPL Loper et al. (2023) mesh renderings are body-tight and do not capture deformations on clothing, like skirts or various hairstyles. To solve this, we learn a second model, that maps from mesh renderings to more realistic images, by controlling the motion with 3D poses. We find out such a simple framework could successfully synthesize realistic and faithful human videos, particularly for long video generations. We show that the 3D control provides a more fine-grained and accurate flow of motion and captures the visual similarities of the imitator faithfully. Because of this, we would like to highlight that 3DHM exhibits much better performance in full-body animation due to its *Texture Mapped* design (Fig. 2), rather than directly using 3D poses as input, as is done in all state-of-the-art methods. Additionally, 3DHM can predict the complete texture map from the opposite side of the human, not just from the front, which none of the previous works can achieve effectively.

To summarize, our 3DHM achieves all the following features simultaneously. First, 3DHM can specify the appearance of the input person and generate human videos with temporal consistency (Fig. 1, 6 and 8). Second, 3DHM generates movements in 3D, rather than 2D (Fig. 2, 4 and 5). Third, the resulting video sequence from 3DHM can be viewed from arbitrary camera viewpoints to visualize the human's movement (Fig. 3 and 7). A detailed comparison with other methods can be found in Section 2 - Related Works. We kindly recommend that reviewers check our anonymous videos at the following link: demo.

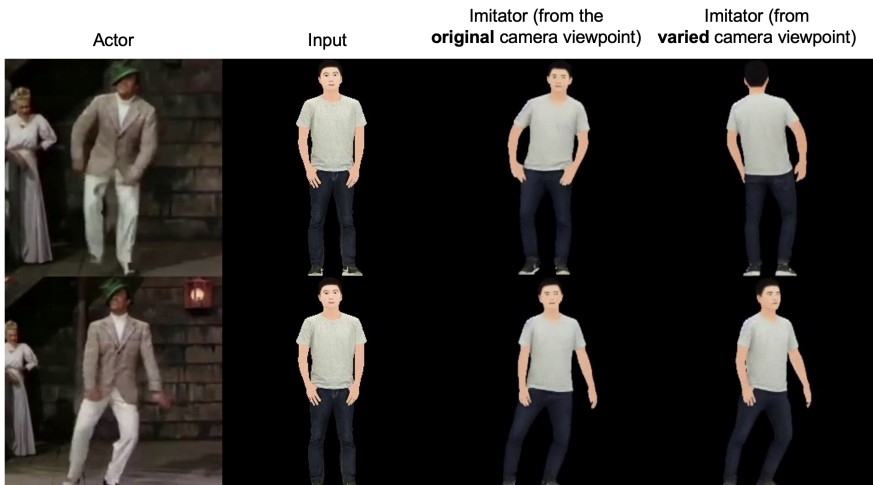

Figure 3: 3DHM can synthesize moving people from both the original camera viewpoint or any other camera viewpoint.

## 2 Related Works

Animating a human from a single input image with various motions is a challenging problem in computer vision. An ideal model should be universal, capable of synthesizing realistic human motions with accurate pose information while maintaining fidelity in appearance with minimal flickering. To ensure flexibility and generalization, the model should support animations from various camera viewpoints. Early methods Chan et al. (2019); Wang et al. (2018) directly learned pose-to-pixel mappings but required separate training for each individual. Make-a-Video (Singer et al., 2022) and Imagen Video (Saharia et al., 2022) could synthesize videos from textual instructions to generate different people; however, their outputs often fail to accurately preserve human properties due to a lack of pose supervision, leading to unnatural compositions. ControlNet (Zhang & Agrawala, 2023) introduced neural network architectures to control large diffusion models with additional input conditions, such as OpenPose (Cao et al., 2017), but was limited to generating single images. GestureDiffuCLIP (Ao et al., 2023) focused on generating co-speech gestures but was not specifically designed for human animation, failing to guarantee realistic human appearance and clothing.

More recently, several works Chang et al. (2023); Wang et al. (2024); Ma et al. (2024); Peng et al. (2024) have developed universal diffusion models for human animation from a single input image and motion guidance, such as Dreampose (Karras et al., 2023), DisCO (Wang et al., 2023), AnimateAnyone Hu (2024), MagicAnimate Xu et al. (2024), and Champ Zhu et al. (2024). DisCO and AnimateAnyone rely on OpenPose (Cao et al., 2017) that provides 2D pose information to animate humans, while Openpose primarily contains the anatomical key points of humans, it can not be used to indicate the body shape, depth, or other related human body information. DreamPose and MagicAnimate leverage DensePose (Güler et al., 2018), which offers a 2.5D representation by mapping 2D pixels to a 3D surface without fully reconstructing the 3D geometry. Although 2.5D poses improve generation quality, they cannot represent all human motions, making the transition to 3D essential. Only a few works have explored this direction, with Champ Zhu et al. (2024) being a notable example, utilizing multiple condition maps rendered from SMPL mesh to enhance control over different viewpoints. However, aligning output pixels for training regularization often overfits these models to specific training data, limiting generalization to novel subjects and different camera angles. Considering both the advancements and limitations of previous works, our proposed 3DHM integrates three key components (as illustrated in Fig. 2): 1) instead of directly inputting poses, we leverage traditional graphical techniques to reconstruct a complete texture map from a single image, ensuring adaptability to any input; 2) we then map the texture map to the 3D pose to obtain a texture-mapped 3D human, enabling realistic synthesis from various camera viewpoints; 3) since the generated frames only include the human figure and may still have incomplete appearances, we further refine the rendered video to match the original input's appearance. We also summarize these properties and compare them with state-of-the-art methods in Table 1.

| Method | Generate Videos | Specify Appearance | Animate-2D | Animate-3D | Specify Camera Viewpoints |
|---|---|---|---|---|---|
| ControlNet (2023) | ✗ | ✓ | ✓ | ✗ | ✗ |
| DreamPose (2023) | ✓ | ✓ | ✓ | ✗ | ✗ |
| DisCO (2024) | ✓ | ✓ | ✓ | ✗ | ✗ |
| Animate Anyone (2024) | ✓ | ✓ | ✓ | ✗ | ✗ |
| Magic Animate (2024) | ✓ | ✓ | ✓ | ✗ | ✗ |
| Champ (2024) | ✓ | ✓ | ✓ | ✓ | ✗ |
| **3DHM (Ours)** | ✓ | ✓ | ✓ | ✓ | ✓ |

Table 1: **Overview of 3DHM's properties** compared with other state-of-the-art methods. Animate-2D refers to synthesizing moving people with various motions in 2D spaces. However, since 2D motion cannot represent all possible movements, Animate-3D involves synthesizing moving people in 3D spaces.

# 3 Synthesizing Moving People

In this section, we discuss our two-stage approach for imitating a motion sequence. Our 3DHM framework embraces the advantage of accurate 3D pose prediction from the state-of-the-art predicting models 4DHumans (Rajasegaran et al., 2022; Goel et al., 2023), which could accurately track human motions and extracts 3D human poses of the actor videos. For any given video of the actor we want to imitate, we use 3D reconstruction-based tracking algorithms to extract 3D mesh sequences of the actor. For the inpainting and rendering part, we rely on the pre-trained Stable Diffusion (Rombach et al., 2022b) model, which is one of the most recent classes of diffusion models that achieve high competitive results over various generative vision tasks.

Our approach 3DHM is composed of two core parts: Inpainting Diffusion for texture map in-painting as Stage-1 and Rendering Diffusion for human rendering as Stage-2. Fig. 2 shows a high-level overview of our framework. In Stage-1, first, for a given single view image, we extract a rough estimate of the texture map by rendering the meshes onto the image and assigning pixels to each visible mesh triangle such that when rendered again it will produce a similar image as the input image. This predicted texture map has only visible parts of the input image. The Stage-1 Diffusion in-painting model takes this partial texture map and generates a complete texture map including the unseen regions. Given this complete texture map, we generate intermediate renderings of SMPL Loper et al. (2023) meshes and use Stage-2 model to project the body-tight renderings to more realistic images with clothing. For the Stage-2 model, we apply 3D control to animate the imitator to copy the actions of the actor.

## 3.1 Texture Map Inpainting

The goal of Stage-1 model is to produce a plausible complete texture map by inpainting the unseen regions of the imitator. We extract a partially visible texture map by first rendering a 3D mesh onto the input image and sample colors for each visible triangle following 4DHumans Goel et al. (2023).

**Input.** We first utilize a common approach to infer pixel-to-surface correspondences to build an incomplete UV texturemap (Xu & Loy, 2021; Casas & Trinidad, 2023) for texturing 3D meshes from a single RGB image. We also compute a visibility mask to indicate which pixels are visible in 3D and which ones are not.

**Target.** We train our model on a large 3D human texture dataset Liu et al. (2024), which contains 50k high-fidelity textured UV map of SMPL Loper et al. (2023). To strengthen the model's 3D geometry consistency in completing the partial texturemap, We densely sample a group of visibility masks from 360 degrees of SMPL mesh, which then mask out Ground-Truth texture map to produce the pseudo-partial texture map during training the inpainting model. Benefiting from the extensive collection of texture maps from diverse human appearances, as well as the numerous visibility masks from various viewpoints.

**Model.** We finetune directly on the Stable Diffusion Inpainting model Rombach et al. (2022a) that shows great performance on image completion tasks. Given a single RGB human image, we predict the human mesh and calculate its corresponding visibility mask and partial texture map, which is then recovered by the in-painting model to complete texture map for the human. We lock the text encoder branch during training

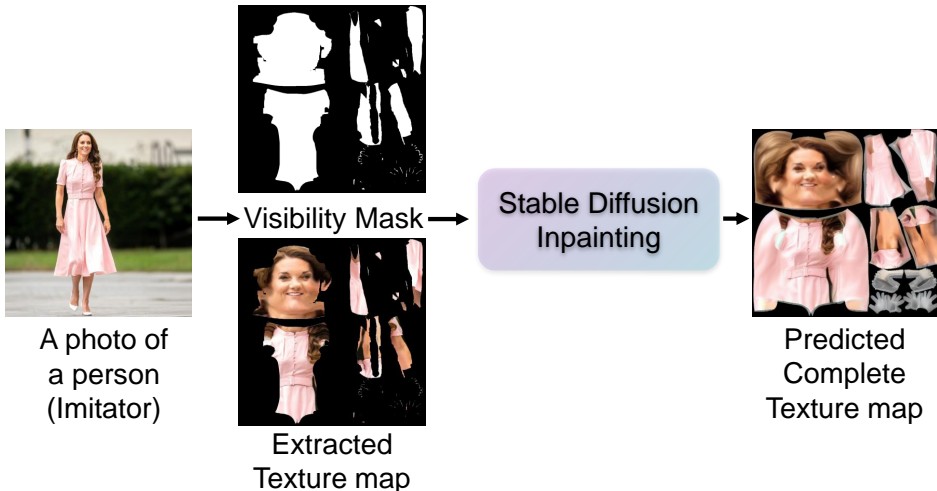

Figure 4: **Stage-1 of 3DHM:** In the first stage, given a single view image of an imitator, we first apply 4Dhumans Goel et al. (2023) style sampling approach to extract partial texture map and its corresponding visibility map. These two inputs are passed to the in-painting diffusion model to generate a plausible complete texture map. In this example, while we only see the **front view** of the imitator, the model was able to hallucinate a plausible back region that is consistent with their clothing.

and feed "3D realistic human, UV texturemap" as input text condition. We refer to our trained model as Inpainting Diffusion. See Fig. 4 for the model architecture.

## 3.2 Human Rendering

In Stage-2, we aim to obtain a realistic rendering of a human imitator doing the actions of the actor. While the intermediate renderings (rendered with the poses from the actor and texture map from Stage-1) can reflect diverse human motion, these SMPL mesh renderings are body-tight and cannot represent realistic rendering with clothing, hairstyles, and body shapes. We train a model for realistic rendering, in a fully self-supervised fashion, by relying on the actor as the imitator. We obtain a sequence of poses from 4DHumans Goel et al. (2023) for each training video and use Stage-1 on single frames to obtain a complete texture map. We then pair the intermediate renderings (i.e. the rendered texture maps on the 3D poses) with the original frames from which they were obtained. We collect a large amount of paired data and train our Stage-2 diffusion model with conditioning.

**Input:** We first apply the generated complete texture map from Stage-1 to the actor's 3D body mesh sequences to obtain the intermediate rendering. Note that the rendering can only reflect the clothing that fits the 3D mesh (body-tight clothing) but fails to reflect the texture outside the SMPL body (e.g., the puffed-up skirt region, or hat). To obtain the human with complete clothing texture, we input the obtained intermediate renderings and the original image of the person into Rendering Diffusion to render the human in a novel pose with a realistic appearance.

**Target:** Since we collected the data by assuming the actor is the imitator, we have the paired data of the intermediate renderings and RGB images. This allows us to train this model on lots of data, without requiring any direct 3D supervision.

**Model.** Similar to ControlNet, we directly clone the weights of the encoder of the Stable Diffusion Rombach et al. (2021) model as our Controllable branch ("trainable copy") to process 3D conditions. We freeze the pre-trained Stable Diffusion. In the meanwhile, we input a texture-mapped 3D human at time $t$ and original human photo input into a fixed VAE encoder and obtain texture-mapped 3D human latents ($64 \times 64$) and appearance latents ($64 \times 64$) as conditioning latents. We feed these two conditioning latents into Rendering Diffusion Controllable branch. The key design principle of this branch is to learn textures from human input and apply them to the texture-mapped 3D human during training through the denoising process. The goal is

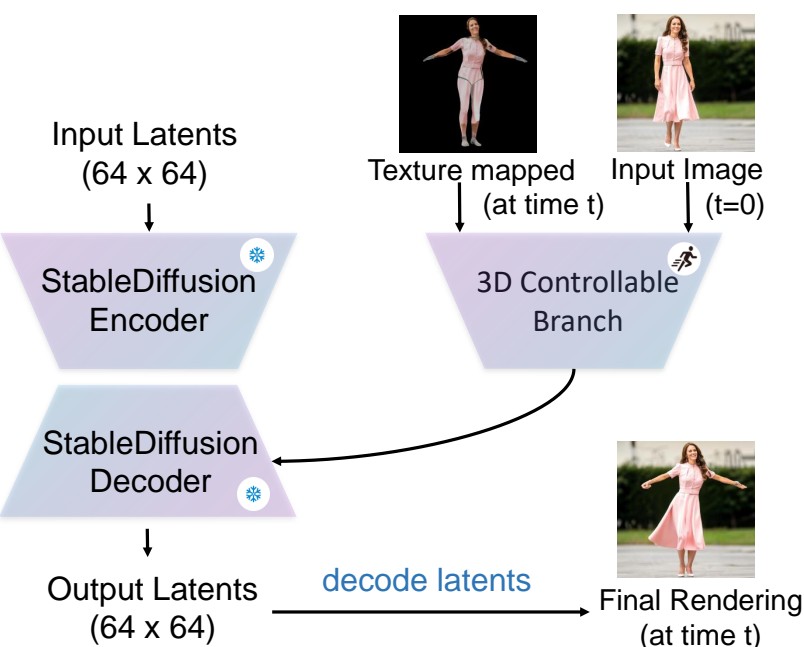

Figure 5: **Stage-2 of 3DHM:** Given an intermediate rendering of the imitator with the pose of the actor and the actual RGB image of the imitator, our model can synthesize realistic renderings of the imitator on the pose of the actor.

to render a real human with vivid textures from the generated(texture-mapped) 3D human from Stage-1. We obtain the output latent and process it to the pixel space via diffusion step procedure and fixed VAE decoder. We refer to our trained model as Rendering Diffusion. In Rendering Diffusion, we predict outputs frame by frame. We show the Stage-2 workflow in Fig. 5.

### 3.3 Experiments

**Dataset.** We collect 2,524 3D human videos from 2K2K (Han et al., 2023), THuman2.0 (Yu et al., 2021) and People-Snapshot (Alldieck et al., 2018) datasets. 2K2K is a large-scale human dataset with 3D human models reconstructed from 2K resolution images. THuman2.0 contains 500 high-quality human scans captured by a dense DLSR rig. People-Snapshot is a smaller human dataset that captures 24 sequences. We convert the 3D human dataset into videos and extract 3D poses from human videos using 4DHumans Goel et al. (2023). We use 2,205 videos for training and other videos for validation and testing. See the Appendix for more details on the dataset distribution on clothing.

**Evaluation Metrics.** We evaluate the quality of generated frames of our method with image-based and video-based metrics. For image-based evaluation, we follow the evaluation protocol of DisCO (Wang et al., 2023) to evaluate the generation quality. We report the average PSNR (Hore & Ziou, 2010), SSIM (Wang et al., 2004), FID (Heusel et al., 2017), LPIPS Zhang et al. (2018), and L1. For video-based evaluation, we use FVD (Unterthiner et al., 2018). For pose evaluating 3D pose accuracy, we use Mean Per-Vertex Position Error (MPVPE) and Procrustes-Aligned Mean Per-Vertex Position Error (PA-MVPVE (Moon et al., 2022)).

**Implementation Details.** We set a learning rate of 5e-05 and use the pre-trained diffusion models for both stages. In Stage-1, we finetune the whole inpainting network. For Stage-2 Rendering Diffusion, we train the Controllable branch and freeze Stable Diffusion backbones. The total number of trainable parameters in this case is 876M. We train Rendering Diffusion for 30 epochs (requires about 2 weeks on 8 NVIDIA A100 GPUs with a batch size of 4). During inference, we only need to run Stage-1 once to reconstruct the full texture map of the imitator, which is used for all other novel poses and viewpoints. In Stage-2, the initial RGB frame of the imitator is conditioned for all frames, to produce samples that are temporally consistent.

| Method | PSNR↑ | SSIM ↑ | FID ↓ | LPIPS ↓ | L1 ↓ | FID-VID↓ | FVD ↓ |
|---|---|---|---|---|---|---|---|
| DreamPose | 35.06 | 0.80 | 245.19 | 0.18 | 2.12e-04 | 113.96 | 950.40 |
| DisCO | 35.38 | 0.81 | 164.34 | 0.15 | 1.44e-04 | 83.91 | 629.18 |
| MagicAnimate | 32.57 | 0.65 | 300.66 | 0.29 | 5.80E-04 | 140.45 | 900.70 |
| **Ours** | **36.18** | **0.86** | **154.75** | **0.12** | **9.88e-05** | **55.40** | **422.38** |

Table 2: **Quantitative comparison on generation quality.** We compare our method with prior works on pose condition generation tasks and measure the generation quality of the samples.

### 3.3.1 Quantitative Results

**Baselines.** We compare our approaches with past and state-of-the-art methods: DisCo (Wang et al., 2023), DreamPose (Karras et al., 2023), MagicAnimate (Xu et al., 2024), and ControlNet (Zhang & Agrawala, 2023) (for pose accuracy comparisons)[1]. We set inference steps as 50 for all the approaches for fair comparisons.

**Comparisons on Frame-level Generation Quality.** We compare 3DHM with other methods on 2K2K test dataset, which is composed of 50 unseen human videos, at $256 \times 256$ resolution. For each human video, we take 30 frames that represent the different viewpoints of each unseen person. The angles range from 0° to 360°, we take one frame every 12° to better evaluate the prediction and generalization ability of each model. As for DisCO, we strictly follow their setting and extract OpenPose for inference. We extract DensePose for inference DreamPose and MagicAnimate. We evaluate the results and calculate the average score over all frames of each video. We set the background as black for all approaches for fair comparisons. We report the average score of the same 50 videos and show the comparisons in Table 2. We observe that 3DHM outperforms all the baselines in different metrics.

**Comparisons on Video-level Generation Quality.** To verify the temporal consistency of 3DHM, we also report the results following the same test set and baseline implementation as in image-level evaluation. Unlike image-level comparisons, we concatenate every consecutive 16 frames to form a sample of each unseen person on challenging viewpoints. The angles range from 150° to 195°, we take one frame every 3° to better evaluate the prediction and generalization ability of each model. We report the average score overall of 50 videos and show the comparisons in Table 2. We observe that 3DHM, though trained and tested by per frame, still embraces significant advantage over prior approaches, indicating superior performance on preserving the temporal consistency with 3D control.

**Running Cost.** Here we outline the comparison of parameters and running time with other methods in Table 2 using a single GPU A100. We show the comparison in Table 3.

**The Benefit of using 3D Control for Pose Accuracy.** To further evaluate the generalization of our model, we estimate 3D poses from generated human videos from different approaches via a state-of-the-art 3D pose estimation model 4DHumans. We use the same dataset setting mentioned above and compare the extracted poses with 3D poses from the target videos. Following the same comparison settings with generation quality, we evaluate the results and calculate the average score over all frames of each video. Beyond DreamPose and DisCO, we also compare with ControlNet, which achieves the state-of-the-art in generating images with conditions, including Openpose control. We input the same prompts as ours 'a real human is acting' and the corresponding Openpose as conditions for ControlNet. We report the average score overall of 50 test videos and show the comparisons in Table 4. We could notice that 3DHM could synthesize moving people following the provided 3D poses with very high accuracy in comparison with other approaches. We assume this is because previous approaches cannot learn all the motions by directly predicting the pose-to-pixel mapping.

---

[1]We utilize the open-source official code and models provided by the authors to implement these baselines. We use diffusers (von Platen et al., 2022) for ControlNet and Openpose extraction, and Detectron2 for DensePose extraction for MagicAnimate and DreamPose. Since Chan et al. (2019) can only work for animating a specific person, we don't compare with it in this paper.

| Method | Time (second/frame) | Parameter |
|---|---|---|
| DreamPose | 22.0 | 1.0B |
| DisCO | 5.0 | 2.0B |
| MagicAnimate | 10.0 | 2.0B |
| Ours | **3.2** | 1.0B |
| Ours (scaled-up) | 8.9 | 2.0B |

Table 3: **Comparison of running cost.** We compare inference time for different models, and we can see that our model is faster in comparison with our models. We also present the time and parameters of our scaled-up model, which achieves enhanced control and consistency while delivering the highest quality. Since it further improves video consistency through temporal layers, it results in a slightly longer processing time compared to the DisCO baseline.

| Method | MPVPE ↓ | PA-MPVPE ↓ |
|---|---|---|
| DreamPose | 123.07 | 82.75 |
| DisCO | 112.12 | 63.33 |
| ControlNet | 108.32 | 59.80 |
| **Ours** | **41.08** | **31.86** |

Table 4: **Quantitative comparison of pose accuracy.** We measure pose accuracy in the generated images by comparing them to the ground truth poses. The results show that our model accurately preserves poses in the generated images.

| Settings | PSNR↑ | SSIM ↑ | FID ↓ | LPIPS ↓ | L1 ↓ | FID-VID↓ | FVD ↓ | MPVPE ↓ | PA-MPVPE ↓ |
|---|---|---|---|---|---|---|---|---|---|
| **3DHM (proposed)** | 36.18 | 0.86 | **154.75** | **0.12** | 9.88e-05 | **55.40** | **422.38** | 41.08 | 31.86 |
| w/o Texture map | 35.00 | **0.78** | 237.42 | 0.20 | 2.35e-04 | 113.97 | 632.67 | 92.94 | 59.18 |
| w/o Appearance Latents | 36.07 | 0.86 | 167.58 | **0.12** | 1.03e-04 | 93.21 | 715.51 | 41.99 | 32.82 |
| adding SMPL parameters | **36.42** | 0.87 | 157.60 | **0.12** | **8.87e-05** | 72.35 | 579.90 | **39.16** | **29.67** |

Table 5: **Ablation study of Rendering Diffusion.** We compare the frame-wise generation quality, video-level generation quality and the pose accuracy under different settings. We notice both texturemap reconstruction and appearance latents are critical to the model performance. The results show that although adding SMPL parameters achieve better performance on frame-level setting but may yield worse temporal consistency than default settings. Note: we use **bold** to represent the best result and underline to represent the second-best result.

### 3.3.2 Ablation Study

To further verify the components of our methods, we train on training dataset and test on test datasets. We extract the 3D rendered pose from these 50 test video tracks. Same with the settings in quantitative comparison, we calculate the average scores among all the generated frames and targeted original frames and report the results on both frame-wise metric (PSNR, SSIM, FID, LPIPS, L1), video-level metric (FID-VID, FVD) and pose accuracy (MPVPE, PA-MPVPE) in Table 5. We find that both texture map reconstruction and appearance latents are critical to the model performance. Also, we notice that directly adding SMPL parameters into the model during training may not bring improved performance considering all evaluation metrics.

## 4 Scale up 3DHM

In this section, we discuss how to scale up our method to real-world domains. We first discuss the challenges of detailed control for Diffusion models and then further explore a lossless approach to ensure the visual

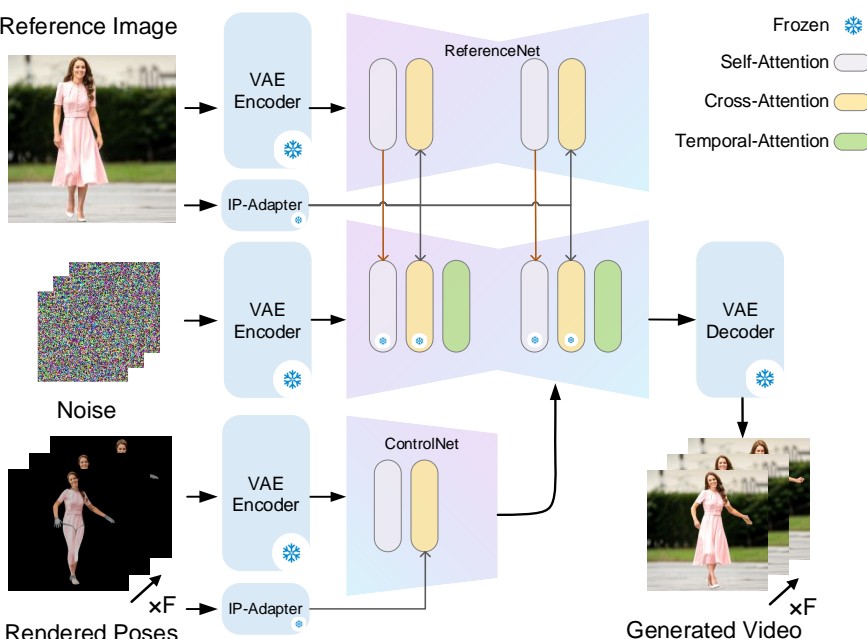

Figure 6: **Scaled up Stage-2 of 3DHM Model:** To enable consistent background and human generation, we train ReferenceNet with ControlNet, and then only finetune the temporal-attention layer of the UNet and keep other components frozen.

consistency of human identity and background from input reference images. To improve video consistency, we utilize a temporal diffusion model (Guo et al., 2024) to learn the temporal correlation within motion sequences. The detailed framework is shown in Fig. 6.

## 4.1  Enhance Appearance Alignment

The key challenge in scaling up our method to real-world domains is to maintain both the complex background and the human appearance from reference images consistently. The Stable Diffusion Model Rombach et al. (2022b), trained for text-to-image tasks, prioritizes semantics over low-level visual details. However, our Stage-2 rendering requires low-level details to fine structural and appearance reconstruction. Therefore, we use a lightweight image adapter Ye et al. (2023) to condition diffusion on image prompts, and add a trainable branch of the Stable Diffusion model, namely Reference Net, to enhance consistency on both input's background and human appearance.

**Input.** Same with Stage-2, we input the refined imitator's texture map from Stage-1 with actor's 3D motions to get intermediate renderings. The intermediate rendering is then fed to the 3D Controllable branch as the motion condition. The original imitator's RGB image is fed to the Reference Net and the image adaptor as the appearance guidance.

**Dataset.** We collect 1,000 real human dancing videos (2 - 10 seconds) from the Internet as an important complement to our 3D virtual dataset.

**Target.** We want to scale up our model training with both collected real videos and 3D virtual datasets together to improve model's generalizability in complex poses and the various 3D views, respectively.

**Model.** To better inject the input image's appearance into the denoising backbone, we make a trainable copy of the pre-trained Stable Diffusion as our ReferenceNet. We now separately extract the imitator's appearance with ReferenceNet and inject them into the backbone by levels to condition the diffusion. This lossless way is essential to keep a consistent background and human appearance for different poses. Besides, a pretrained IP-adapter Ye et al. (2023) is also integrated into cross-attention layers to better control the human identity. From the predicted SMPL parameters, we can further align body shapes between the input image and motion sequences.

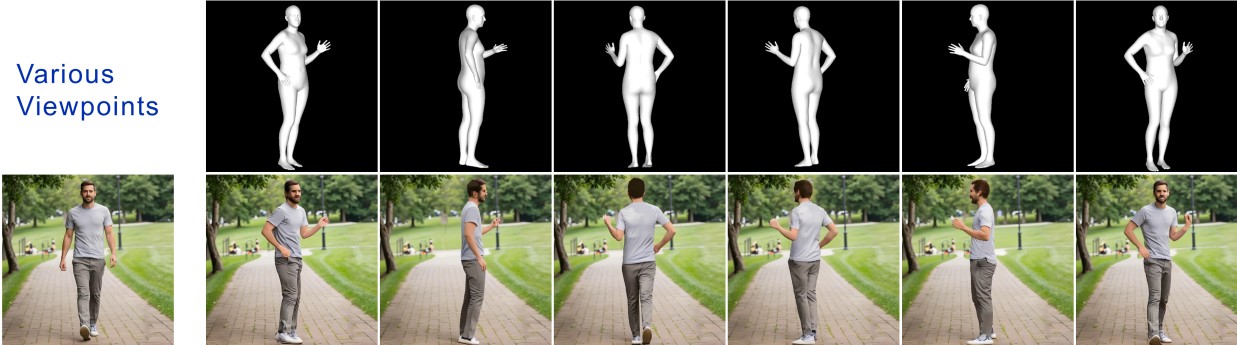

(a) 3DHM with random 3D poses from various viewpoints. We show that even if the person's photo is from a specific angle, our Stage-1 can help reconstruct the full texture map to obtain full body information. Stage-2 can better align the appearance based on a given input.

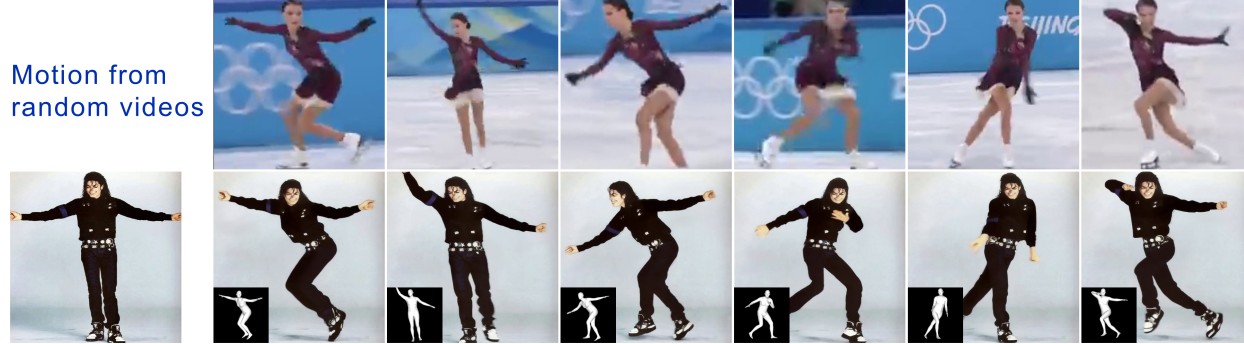

(b) 3DHM with motions from random YouTube Videos.

Figure 7: Qualitative results on different viewpoints of the same pose and motions from random videos.

## 4.2 Temporal Consistency

With the strong image-based guidance mentioned above, our model can generate a video frame by frame. However, the generated frames may suffer from jittering due to the lack of temporal consistency. In our scaled-up model, we insert temporal layers pre-trained on a large video dataset Guo et al. (2024) to improve the motion coherence and appearance consistency. Previous works Wang et al. (2023); Karras et al. (2023); Hu (2024) also have similar layers and success for short video generation. However, these methods can hardly generate consistent long videos. The sliding windows strategy they use will cause instability and randomness between each short video clip generated. Based on that, our model further guides long video generation by conditioning on previously generated frames.

**Input.** During training, for each video clip, we extract $F$ consecutive frames as the target of actor's motion sequence and randomly pick a frame as the imitator's reference image. Now the 3D Control branch takes $F$ consecutive intermediate rendering to animate the imitator's image and generate a $F$ frames video. Since the model only extracts the input image's latent once from Reference Net for each video clip, it almost cost no extra computing time.

**Model.** We define the short video clip as $V \in \mathbb{R}^{B \times C \times F \times H \times W}$, with batch size $B$, the number of channels $C$, the number of consecutive frames $F$, height $H$ and width $W$ respectively. The temporal layers are inserted at each resolution level. For each level $i$ , the 5D latents $v_i \in \mathbb{R}^{b \times c \times f \times h \times w}$ is reshaped to $(b \times h \times w) \times f \times c$ within the temporal layer as self-attention to align feature maps across frames. However, during long video generation, since the video clips are generated independently and concatenated together, the different random noises will cause inconsistency between each video clip. To facilitate the cross-clips consistency, we take the last frame $V_k^f$ from $k$-th generated video clip $V_k^{1:f}$ to condition on the next video clip generation $V_{k+1}^{1:f}$. The $V_k^f$ is input to the Reference Net to extract corresponding latents for each resolution level $i$, and then

| Method | PSNR↑ | SSIM ↑ | LPIPS ↓ | L1 ↓ | FID-VID↓ | FVD ↓ |
|---|---|---|---|---|---|---|
| DreamPose | 28.04 | 0.509 | 0.450 | 6.91E-04 | 80.51 | 551.56 |
| DisCO | 29.03 | 0.668 | 0.292 | 3.78E-04 | 59.90 | 292.80 |
| Animate Anyone | 29.56 | 0.718 | 0.285 | - | - | 171.9 |
| MagicAnimate | 29.16 | 0.714 | 0.239 | 3.13E-04 | 21.75 | 179.07 |
| Champ | **29.84** | 0.773 | 0.235 | 3.02E-04 | 26.14 | **170.20** |
| **Ours (Scaled-up)** | 29.79 | **0.785** | **0.231** | **2.93E-04** | **20.68** | 176.28 |

Table 6: **Quantitative comparison on TikTok Dataset.** We compare our scaled-up model with the previous methods. We would like to highlight that 3DHM exhibits much better generalization due to its *Texture Mapped* design, rather than directly using 3D poses as input like all other methods.

concatenated with the following 5D video latents along the temporal dimension. The conditioned temporal layers at level $i$ now attention across latent $v_i^{0:f} \in \mathbb{R}^{b \times c \times (1+f) \times h \times w}$ and then truncate the previous frame $v_i^0$ to get the conditioned results $\bar{v}_i^{1:f}$. Zero-initialize Zhang & Agrawala (2023) is also applied to the temporal layers to eliminate harmful noise during training.

### 4.3 Experiments

**Evaluation Metrics.** We evaluate the quality of generated videos from our scaled-up model. with the image-based metrics, including the average PSNR, SSIM, LPIPS, and L1. For video-based evaluation, we use FVD and FID-VID.

**Implementation Details.** We initialize the scaled-up model with the Stable Diffusion model and the temporal layers from AnimateDiff (Guo et al., 2024). The temporal sliding window size is set to 16 frames for both training and inference. We randomly sample the video clips from both virtual and real datasets and train the scaled-up stage-2 model for 10 epochs.

#### 4.3.1 Quantitative Results

TikTok dataset (Jafarian & Park, 2021) is a common benchmark for human video generation tasks. We compare our scaled-up model's performance with previous state-of-the-art methods in Table 6. For a fair comparison with previous methods, we fine-tune our model and evaluate it on the TikTok dataset. The quantitative results show that our scaled-up model has better performance for both image and video quality.

#### 4.3.2 Qualitative Results

With the aid of 3D assistance, our approach has the potential to produce human motion videos in various scenarios. We consider challenging 3D poses and motions from 2 sources: 3D human videos and random YouTube videos.

**Poses from Unseen 3D Human Videos.** We test our scaled-up model on different 3D human poses and viewpoints from the 2K2K dataset. We verify that the tested video has never appeared in training data. We show the results in Fig. 7a.

**Motions from Random YouTube Videos.** We test our model on very different motions from randomly downloaded YouTube videos of an unseen human. We display the results in Fig. 7b. The results show that 3DHM can efficiently animate any person using random motion resources, accurately following the 3D poses from challenging motion sources.

We also compare the results of the official model from state-of-the-art methods on random real human photos which ensures distinct data distribution. We display the qualitative results of various poses on real human photos in Fig. 8. We notice that 3DHM can generalize well to unseen real humans and motions though it is only trained with limited data. Since DreamPose requires subject-specific finetuning of the UNet to achieve

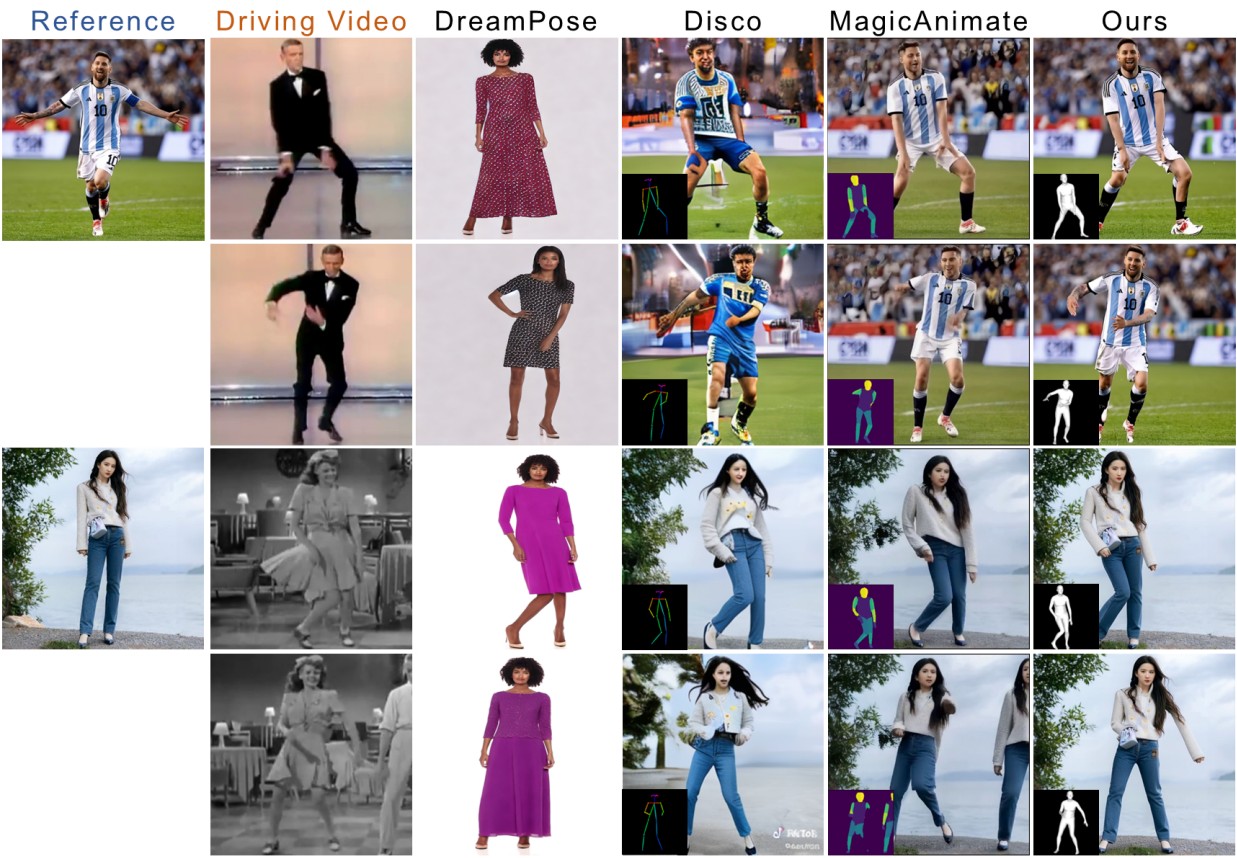

Figure 8: **Qualitative comparison of real human photos** with state-of-the-art methods. Please zoom in to examine the details.

better results, it cannot directly generalize well on a random human photo. As for DisCO, though it has been pre-trained on multiple public datasets for better generalizability, it still fails to synthesize people without the target pose. MagicAnimate uses 3D pose features (DensePose) which better controls the appearance of input images but always suffers from severe artifacts on DensePose segmentation maps, which severely ruins the pose accuracy and consistency. Compared to previous methods, 3DHM gets improvement from adding rigid 3D control to better correlate the appearance to the poses and preserve the body shape. In contrast, conditioning on OpenPose or DensePose cannot guarantee the mapping between textures and poses, which undermines the models's generalization ability.

### 4.3.3 Limitations

As 3DHM has been trained with limited data (around 2K synthetic humans and 1K real humans), it might struggle to predict the texture details of the unseen side of the input human photo. However, we believe this issue can be mitigated by scaling up with more human data.

## 5 Conclusion

In this paper, we propose 3DHM, a two-stage diffusion model-based method that enables human animation from a single random photo and a target sequence of human poses. A notable aspect of our approach is the use of a cutting-edge 3D pose estimation model to generate 3D poses, combined with classical graphics techniques to synthesize people with arbitrary poses, allowing our model to be trained on any video. Our method is well-suited for long-range motion generation and can handle arbitrary poses with superior performance compared to previous approaches. It preserves the target motion's poses, as well as clothing and facial identities, while ensuring smoother transitions between frames.

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
