# OpenReview forum: "Synthesizing Moving People with 3D Control"
_TMLR — Rejected by TMLR_

### Review · Reviewer_rPJk · 2025-08-04

**Summary Of Contributions:**

**[A Quick Summary]** This paper introduces 3D Human Motion (3DHM) diffusion, a two-stage generative pipeline that turns a single portrait photo into a realistic video of the same person performing arbitrary 3-D poses with different camera views.  In the first stage, an in-painting diffusion model hallucinates a full UV texture map of the subject; in the second, a pose-conditioned diffusion renderer synthesises each frame while maintaining identity, lighting, and background consistency.  A scale-up variant with a Reference-Net and temporal attention further stabilises long sequences.  Extensive experiments on a new 2 524-video synthetic dataset and 1 000 in-the-wild TikTok clips show state-of-the-art FID/FVD and pose accuracy, with ablations confirming the value of each design choice and a runtime of ~3 s per frame on a single GPU.

**[Contributions] They are listed down as follow.**
- A two stage 3DHM diffusion pipelines that in-painting a full UV texture from one partial image and rendering video frames with SMPL control.
- A scale-up method that improves the stability for lengthy animations.
- The pipeline is more efficient than other methods, with ~3 seconds per frame while matching or even surpassing the quality of previous methods.
- A new dataset contains 2524-video, 1000 TikTok clips and other metrics.

**Audience:**

Yes

**Audience Explanation:**

TMLR readers who focus on applied generative modelling, controllable video synthesis, and 3-D-aware diffusion architectures will find this work particularly compelling, even though its emphasis is more on engineering innovation than on new ML theory (the question here is **at least** some individuals).  Releasing the dataset, code, and pretrained weights would further widen its appeal, especially to the engineering side who value reproducibility and benchmarks.

**Broader Impact Concerns:**

A Broader-Impact Statement is recommended. It may contain follow points:

- safeguards against impersonation and malicious use.
- an informed-consent or watermarking strategy for publicly released models.

The reviewer believe adding such a section would align the work with TMLR’s ethics guidelines and help reviewers gauge societal impact.

**Claims And Evidence:**

Yes

**Claims Explanation:**

Most of the authors’ claims are convincingly supported, but a few contributions would benefit from stronger or more complete evidence:

- An additional experiment on the effects of the Reference-Net and the temporal-attention layers would clarify how each component improves long-sequence stability.
-  Reporting key runtime settings—especially batch size, resolution, and GPU memory usage—for both 3DHM and the baselines would make the speed comparisons more transparent and reproducible.
- A small user study (or a face-matching metric) evaluating how well the disentangled 3-D control maintains identity across novel views and motions would further benefit this claim.

Including these results or parameters is encouraged, but their absence does not undermine the paper’s primary contributions, which remain well validated.

**Requested Changes:**

I consider the paper slightly above the acceptance threshold; the following changes would further strengthen it:

- Provide complete experimental details for each ablation. In particular, explain how performance is measured when texture mapping is removed—at first glance the framework should not function at all without texture information.

- The demos reveal noticeable inconsistencies in texture-rich regions (e.g., faces and hands). A brief analysis of when and why these artifacts occur would benefit readers.

- Add experiments that isolate and quantify the contributions of the Reference-Net and temporal-attention modules.

- Include an Ethics & Broader-Impact statement

- Complement quantitative results with a user study or automated face-verification metric, not just showing the qualitative results.

---

### Review · Reviewer_X8d3 · 2025-08-06

**Summary Of Contributions:**

The paper proposes a diffusion model-based approach to animate a person from a single input image using a given target 3D motion sequence. The method is built on two main components: Learning Priors for Invisible Parts, Diffusion-Based Rendering Pipeline. The explicit 3D control also enables rendering the animated person from various synthetic camera trajectories. Experiments demonstrate the method's robustness in generating prolonged motions and handling a wide range of challenging and complex poses compared to previous approaches.

# Strengths

- Separating the problem into learning invisible parts and rendering novel poses makes the approach robust and modular, contributing to high-quality results.


- The paper highlights the method's resilience in generating prolonged and complex motions and varied challenging poses, indicating its versatility in diverse scenarios.

- The paper is well-written and easy to follow.

**Audience:**

Yes

**Audience Explanation:**

The task is very interesting and based on diffusion models. It will be more attractive, and the submission supports the camera control during the animation, which is an important feature during the animation.

**Claims And Evidence:**

No

**Claims Explanation:**

While generally visually similar, maintaining perfect fidelity to very fine identity details (e.g., specific facial nuances, subtle clothing textures) across extreme poses and viewpoints from a single input image can be inherently challenging. How to ensure the consistency of human identities.

- The training data is so small, how can to generalize to an extremely wide variety of clothing styles, body shapes, or accessories that may be underrepresented in training data?

- Beyond basic clothing and texture, does the model allow for fine-grained control over specific material properties, lighting conditions, or more nuanced facial expressions/details on the animated person? How does the method handle the interaction of the animated person with a complex 3D environment, including casting and receiving shadows, reflections, and accurate occlusions? Is the person generated in isolation, or can they be seamlessly integrated into a scene?

- How sensitive is the method to variations in the quality (e.g., resolution, lighting, background clutter, occlusions) of the single input image? Are there specific recommendations for optimal input image characteristics?

- While temporal consistency is claimed, what specific mechanisms within the diffusion process are crucial for ensuring high temporal smoothness and preventing subtle lags or visual jumps, particularly in very long animation sequences?

- Very Important point, the paper focuses on human animation, but there are no video-based results and comparisons. It is a strong weakness in the evaluations. Although the paper has achieved better results according to the tables and figures, the video-based results are more important. I strongly encourage the authors to add a video to present more results and comparisons, which should make it more convincing.

- Some similar works: https://github.com/ali-vilab/UniAnimate-DiT, https://arxiv.org/pdf/2504.14899, some discussion and comparison should be considered. BTW, the second one also supports the camera control.

**Requested Changes:**

See weaknesses.

---

### Review · Reviewer_1Nyf · 2025-08-08

**Summary Of Contributions:**

The authors propose a method to animate a person in an image with a SMPL motion sequence. Their approach consists of two primary steps. In the first, the authors perform UV projection of the target image onto an estimated SMPL body mesh, followed by UV-space texture inpainting (*Inpainting Diffusion*). Second, a ControlNet (*Rendering Diffusion*) is conditioned on a render of the texture-inpainted body mesh in a pose, to produce an image of the target in that specified pose. The authors also introduce a temporal adapter (*Scale up 3DHM*).

**Additional Comments:**

1. Does the self-supervision approach assume the background is static? There seems to be a bit of background movement in the Santa Claus video.
2. Why is the background set to black for "fair comparison"? Might this be out-of-distribution to the baseline models?
   > We set the background as black for all approaches for fair comparisons.

   Is the authors' method specifically trained on images with black backgrounds?
3. Could the authors elaborate on the following? How are parameters "added"?
   > Also, we notice that directly adding SMPL parameters into the model during training may not bring improved performance considering all evaluation metrics.
4. Were the baselines similarly finetuned?
   > For a fair comparison with previous methods, we fine-tune our model and evaluate it on the TikTok dataset.
5. The authors write:
   > DreamPose and MagicAnimate leverage DensePose (Güler et al., 2018), which offers a 2.5D representation by mapping 2D pixels to a 3D surface without fully reconstructing the 3D geometry. Although 2.5D poses improve generation quality, they cannot represent all human motions, making the transition to 3D essential.

   However, their method does not appear to be actually "3D." How might a 2D render be considered 3D control? Are there not infinitely many 3D pose--texture combinations that could produce a given 2D render? Unlike Champ, which incorporates:
   > rendered depth images, normal maps, and semantic maps obtained from SMPL sequences, alongside skeleton-based motion guidance

	it does not seem correct to characterize 3DHM as 3D. How might they authors rectify this? If control were given directly as 3D body parameters, it would be unambiguously 3D, but rendering (projection) removes what differentiated DensePose from SMPL.

**Audience:**

Yes

**Audience Explanation:**

Yes. The topic is interesting and relevant to some portion of TMLR's audience. The self-supervised approach for training the "Rendering Diffusion" model is creative and seems to work fairly well.

**Broader Impact Concerns:**

The authors edited the supplementary materials in the past day to include a "Broader-Impact Statement" section.

The authors write:
> We strongly recommend that any future use of this technology involving real individuals, especially in datasets, demonstrations, or downstream applications, should be conducted only with explicit, informed consent from the individuals depicted. Researchers and practitioners should refrain from applying the model to identifiable personal images without permission.

Would the authors please clarify whether they have conducted their experiments with "explicit, informed consent from the individuals depicted"?

**Claims And Evidence:**

No

**Claims Explanation:**

1. This is a video-generation paper, but the authors provide no video comparisons. There are also not any image samples shown from Champ (Zhu et al., 2024), the best-performing baseline method employed. It is difficult to evaluate performance claims without the relevant comparisons on what is a very qualitative task.
2. It seems that the input to the authors' method doubles as the ground truth. The authors write:
   > We convert the 3D human dataset into videos and extract 3D poses from human videos using 4DHumans Goel et al. (2023). We use 2,205 videos for training and other videos for validation and testing.

   If this were provided also to the baseline methods, it would be of little consequence. However, the authors seem to apply separate estimators for each:
   > We utilize the open-source official code and models provided by the authors to implement these baselines. We use diffusers (von Platen et al., 2022) for ControlNet and Openpose extraction, and Detectron2 for DensePose extraction for MagicAnimate and DreamPose.

   These keypoint and DensePose annotations should have instead been derived from the SMPL sequences. It is otherwise not possible to rule out the possibility that method performance is dominated by estimator performance. The samples shown in Figure 8 seem to display failure cases of the baseline estimators, suggesting the evaluation comes down to estimator.
3. The authors claim to compare against "state-of-the-art" approaches, but do not include methods from the past year, such as [1] or [2].
4. Detailed in the Requested Changes section are a number of additional issues relating to claims.

[1] Tu, S., Xing, Z., Han, X., Cheng, Z. Q., Dai, Q., Luo, C., & Wu, Z. (2025). StableAnimator: High-quality identity-preserving human image animation. In CVPR (pp. 21096-21106).

[2] Chang, D., Xu, H., Xie, Y., Gao, Y., Kuang, Z., Cai, S., ... & Soleymani, M. (2025). X-Dyna: Expressive dynamic human image animation. In *CVPR* (pp. 5499-5509).

**Requested Changes:**

### Claims and Correctness
1. The baseline methods should be run with their default generation settings, not modified:
   > We set inference steps as 50 for all the approaches for fair comparisons.

	This does not make sense when the baselines use different models in different ways.
2. This was not demonstrated in the paper (either for this method or prior work) and the claim should be removed:
    > Additionally, 3DHM can predict the complete texture map from the opposite side of the human, not just from the front, which none of the previous works can achieve effectively.
3. None of the changes in the "scaled-up" model were ablated, and it was not evaluated against the authors' other model. The authors should not claim the changes are improvements without specifically evaluating them:
    > The quantitative results show that our scaled-up model has better performance for both image and video quality.
4. This was not evaluated:
   > We train this model on texture map space, which makes it more sample-efficient
5. It does not seem that any examples of synthetic camera trajectories were included:
   > In addition to that, the 3D control allows various synthetic camera trajectories to render a person.

   There are also not any background-synthesis results, so it does not appear this is possible with the approach.
6. The authors have not demonstrated that their method works on "any single image of a new person." They claim:
   > the resulting video sequence from 3DHM can be viewed from arbitrary camera viewpoints to visualize the human’s movement

   This has not been demonstrated. The viewpoints shown are all normalized with the individual upright and approximately filling the frame. No zoomed-in shots, camera tilt, top-down, bottom-up, etc. were shown. The claims should be removed. This includes also:
   > instead of directly inputting poses, we leverage traditional graphical techniques to reconstruct a complete texture map from a single image, ensuring adaptability to any input;
7. This was not demonstrated:
   > We show that the 3D control provides a more fine-grained and accurate flow of motion and captures the visual similarities of the imitator faithfully. Because of this, we would like to highlight that 3DHM exhibits much better performance in full-body animation due to its Texture Mapped design (Fig. 2), rather than directly using 3D poses as input, as is done in all state-of-the-art methods.

   To claim that specific architectural differences were the cause of the improvement, the change should be explicitly ablated, with all data and other settings held constant to remove confounds. It's otherwise not possible to determine this.
### Miscellaneous
1. The method name, "3D Human Motions" is not descriptive of the work. The authors are suggested to rename it to something uniquely identifying.
2. ControlNet is not state-of-the-art. The text should be removed:
   > we also compare with ControlNet, which achieves the state-of-the-art in generating images with conditions, including Openpose control.
3. Table 3 should include all methods evaluated.
4. It doesn't seem that the stage-1 "Inpainting Diffusion" model should be characterized as pose-invariant, as the inpainting mask is clearly pose-dependent:
    > We train this model on texture map space, which makes it more sample-efficient since it is invariant to pose and viewpoint.

   The claim should likely be removed.
5. 4DHumans is not "cutting-edge" or a state-of-the-art pose-estimation method, and has not been for years [3]. The claim should be removed throughout:
    > We use state-of-the-art 3D human pose recovery model 4DHumans
### Style and Typos
1. "Rendering Diffusion" is referenced before it is defined.
2. "PA-MVPVE" -> "PA-MPVPE"
3. In TMLR style [4], captions should go above the tables.
4. The Appendix should be included in the main document; see "Format" [5].
5. The bolding is inverted in the SSIM column of Table 5.
6. Section and figure headings are not consistently capitalized.
7. For clarity, `\citep` should be used when the citation is not the object of the text:
    > on top of the SMPL Loper et al. (2023) mesh
8. The reference to "Fig.  1" has an extra space.
9. To improve accessibility, Figures 1, 2, 3, 7, and 8 should be made vectorized like Figures 4, 5, and 6.
10. Caption titles are not consistently capitalized.
11. Preprints that have since been published should be updated with the correct venue information.
12. There is a stray capital in "texturemap, We."

[3] Shin, S., Kim, J., Halilaj, E., & Black, M. J. (2024). WHAM: Reconstructing world-grounded humans with accurate 3D motion. In _CVPR_ (pp. 2070-2080).

[4] https://github.com/JmlrOrg/tmlr-style-file/blob/7bf90efe3a0debbba703c05c43f3ff7e4d4a2992/main.tex#L195

[5] https://jmlr.org/tmlr/author-guide.html

---

> ### Author Response · Authors · 2025-08-12
> **Response to Reviewer 1Nyf [1/2]**
>
> We would like to thank Reviewer 1Nyf for taking the time to review our paper and provide valuable feedback. We are pleased that the reviewer found 3DHM “for training the Rendering Diffusion model to be creative”. We are happy to have the opportunity to address the reviewer’s questions and concerns.
>
> Weaknesses:
>
> **Q1. More qualitative results.**
>
> Thank you for this great suggestion! The reason we didn’t add Champ images is that its performance is similar to MagicAnimate, and we have included the quantitative comparison in Table 6. And we fully agree with your point. We will add the comparison with Champ in our final version.
>
> **Q2. Different estimators.**
>
> There might be some misunderstandings. Please allow us to explain in detail. For these baseline methods, the selection of the estimator is part of their contributions in the paper. Therefore, to reproduce their results, we strictly follow the settings of these methods, including the choice of estimator. Similarly, one of the core contributions of 3DHM is our two-stage design, in which we predict the full texture map and project it onto the 3D meshes instead of simply feeding in the 3D poses. The “texture-mapped” result can then be used as input to provide a prior for Rendering Diffusion.
>
> **Q3. Related works.**
>
> Thanks for pointing out these works. We admit that we were not aware of them, and we will add a comparison with them and cite them in our final version.
>
> **Q4. Requested Changes.**
>
> 1: Thanks for this great suggestion! Please allow us to explain: for most settings, the inference step is set to 50 or fewer, and generally, more inference steps lead to better performance. To ensure a fair comparison, we kept the number of inference steps the same across methods. Based on your comments, we also retested the performance of each method in Table 2. The performance of other methods is slightly lower or similar to the numbers we originally reported.
>
> 2: Thanks for this great suggestion! As far as we know, we are not aware of any prior work that has shown this level of performance. However, we will follow your instruction and remove this claim in our final version.
>
> 3: We apologize for the confusing statement. What we meant is that we proposed a way to scale up 3DHM to achieve better performance, allowing it to compete with other methods that have been trained on much larger datasets, such as MagicAnimate. We will rephrase this statement in our final version.
>
> 4 & 5 & 6: We apologize for the confusing point. Based on the design of 3DHM, we assume these functionalities could be activated using our resilient approach. While these capabilities are not the main focus of our paper, we did not provide additional evaluation or proof to support these statements. We will rephrase these statements in our final version based on your valuable comments.
>
> 7: We fully agree with your points! Therefore, we conducted ablation studies in Section 3.3.2 and Table 5. In our ablation study, we strictly controlled all data and other settings to avoid confusion. We will also rephrase our claim to improve clarity based on your instructions.
>
> **Q5. Miscellaneous.**
>
> 1: Yes, we agree with this point. At the time of submission, we named it this way considering a broad audience. We may consider using "3D human poses" or other terminology after careful sanity checks in our final version.
>
> 2: Yes, we agree with this point. The term “state-of-the-art” might not be appropriate for ControlNet under several evaluation settings, although its performance is still robust and generalizable even compared to the most recent methods. We will rephrase and remove this terminology in our final version.
>
> 3: Yes, we agree with your point. In the original submission, Table 3 aims to show the benefit of using 3D control for pose accuracy. Therefore, we compare methods with different types of poses, including OpenPose (2D pose) and DensePose (2.5D pose).
>
> 4: Yes, this point aligns with the one in Q4-4. We will rephrase this statement in our final version.
>
> 5: Thanks for pointing this out! We will rephrase this statement based on your instructions. We would also like to highlight that 3DHM can be used with different pose estimation methods; the whole framework is resilient to new state-of-the-art methods.
>
> **Q6. Style and Typos.**
>
> We sincerely appreciate your help in improving our paper by pointing out typos and style issues. We will revise these points in our final version based on your instructions.
>
>
> **Q7. Broader Impact.**
> Thanks for this question! Regarding the experiments and figures in the paper, we either use figures from previous methods or publicly available figures, consistent with what prior works have done. Following your instructions, we will also include a note stating that these figures are used solely for research purposes and will not be used for any commercial purposes.

---

> ### Author Response · Authors · 2025-08-12
> **Response to Reviewer 1Nyf [2/2]**
>
> **Q8. Additional Comments.**
>
> 1: Great point! We did not make the assumption that “the background is static,” as our training data includes videos with moving backgrounds.
>
> 2: We made this statement because we aimed to compare the results based solely on the animated human, rather than the animated human plus the background. Since evaluation metrics such as LPIPS and L1 measure global information, we wanted to minimize the background’s influence on the evaluation results. Additionally, we have included a comparison with complex backgrounds in Table 6, which we hope addresses your concern.
>
> 3: Great question! We incorporate the SMPL parameters as an additional input channel to the model.
>
> 4: In this setting, we fine-tune our model using the same data that the baseline model was trained on. We did not fine-tune the baseline model, as it had already been trained with this data.
>
> 5: Thank you for your valuable comments! We will rephrase this statement in the final version of our paper.
>
> Thank you again for your valuable suggestions to strengthen 3DHM! We hope we have addressed your concerns satisfactorily. Please let us know if you have any further concerns or additional questions we can assist with.

---

> > ### Comment · Reviewer_1Nyf · 2025-08-13
> >
> > The authors mention "final version" ten times in their rebuttal. They are reminded that TMLR is a journal, and that it is now the period for revision.
> >
> > Outstanding issues relating to claims and correctness will be reraised following the submission of revisions.

---

> > > ### Comment · Reviewer_1Nyf · 2025-08-26
> > >
> > > It has been two weeks since the authors' last reply. Could they clarify whether they intend to submit revisions or respond to the other reviewers?

---

### Decision · Action_Editor_ixSX · 2025-09-16

**Recommendation:** Reject

**Additional Comments:**

This paper proposes a diffusion-model-based approach to animate a person from a single input image using a given target 3D motion sequence. The method is built on two main components: learning priors for invisible parts, and diffusion-based rendering pipeline. After author response, it received 2 Leaning Reject and 1 Reject recommendations.

The review process revealed reviewers' concerns about the authors’ engagement and responsiveness. All reviewers noted that important issues regarding claims, correctness, and reviewer concerns remain unresolved. The authors have not provided sufficient evidence of revisions, and failed to engage meaningfully with all reviewers. In particular, several responses deferred fixes to a “final version,” but the actual fixes are not provided in the revision. Overall, the lack of adequate response, insufficient revisions, and unaddressed reviewer feedback prevent the AC from confidently assessing the correctness and readiness of the work. At this stage, the paper is not suitable for acceptance. However, if the authors are able to thoroughly revise the manuscript and engage fully with the reviewers, a resubmission could be considered.

**Audience:**

Yes

**Audience Explanation:**

Yes. The topic is interesting and relevant to some portion of TMLR's audience.

**Claims And Evidence:**

No

**Claims Explanation:**

Reviewers provided detailed questions and concerns regarding the paper. However, the authors' response is in general lightweight, and many concerns are therefore left unaddressed, making the claims made in the submission not well supported by convincing evidence. Specifically,

1. The authors need to provide video qualitative comparisons in supplementary material, given the paper focuses on human animation.

2. The authors need to compare against "state-of-the-art" approaches when claiming SOTA, such as methods from the past year.

3. The authors need to address Reviewer 1Nyf's comments regarding claims and correctness in more detail, and sufficiently revise the paper to reflect the promised changes.

4. The authors need to address the generalization concern given the training data is so small.

5. The authors need to add missing related works as mentioned by reviewers.

**Resubmission Of Major Revision:**

The authors may consider submitting a major revision at a later time.